# Atom-to-Device Simulation of MoO_3_/Si Heterojunction Solar Cell

**DOI:** 10.3390/nano12234240

**Published:** 2022-11-28

**Authors:** Jasurbek Gulomov, Oussama Accouche, Zaher Al Barakeh, Rayimjon Aliev, Irodakhon Gulomova, Bilel Neji

**Affiliations:** 1Renewable Energy Sources Laboratory, Andijan State University, Andijan 170100, Uzbekistan; 2College of Engineering and Technology, American University of the Middle East, Egaila 54200, Kuwait

**Keywords:** heterojunction, MoO_3_, simulation, DFT, TCAD

## Abstract

Metal oxides are commonly used in optoelectronic devices due to their transparency and excellent electrical conductivity. Based on its physical properties, each metal oxide serves as the foundation for a unique device. In this study, we opt to determine and assess the physical properties of MoO_3_ metal oxide. Accordingly, the optical and electronic parameters of MoO_3_ are evaluated using DFT (Density Functional Theory), and PBE and HSE06 functionals were mainly used in the calculation. It was found that the band structure of MoO_3_ calculated using PBE and HSE06 exhibited indirect semiconductor properties with the same line quality. Its band gap was 3.027 eV in HSE06 and 2.12 eV in PBE. Electrons and holes had effective masses and mobilities of 0.06673, −0.10084, 3811.11 cm^2^V^−1^s^−1^ and 1630.39 cm^2^V^−1^s^−1^, respectively. In addition, the simulation determined the dependence of the real and imaginary components of the complex refractive index and permittivity of MoO_3_ on the wavelength of light, and a value of 58 corresponds to the relative permittivity. MoO_3_ has a refractive index of between 1.5 and 3 in the visible spectrum, which can therefore be used as an anti-reflection layer for solar cells made from silicon. In addition, based on the semiconducting properties of MoO_3_, it was estimated that it could serve as an emitter layer for a solar cell containing silicon. In this work, we calculated the photoelectric parameters of the MoO3/Si heterojunction solar cell using Sentaurus TCAD (Technology Computing Aided Design). According to the obtained results, the efficiency of the MoO_3_/Si solar cell with a MoO_3_ layer thickness of 100 nm and a Si layer thickness of 9 nm is 8.8%, which is 1.24% greater than the efficiency of a homojunction silicon-based solar cell of the same size. The greatest short-circuit current for a MoO_3_/Si heterojunction solar cell was observed at a MoO_3_ layer thickness of 60 nm, which was determined by studying the dependency of the heterojunction short-circuit current on the thickness of the MoO_3_ layer.

## 1. Introduction

Modern industries utilize metal oxides abundantly due to their semiconducting nature since they have proven useful in a wide variety of electronic and optoelectronic devices including sensors, transducers [1], and solar cells [2]. Metal oxides can be divided into two types depending on their electrical conductivity: n-type and p-type. Due to the rapid popularity of metal oxides, they can be synthesized using the simple Sol–Gel method [3] while mechanically forming a thin layer.

Metal oxides are a key building block for solar cells and photovoltaic devices. They are also utilized to make transparent solar cells [4], e.g., TiO_2_/NiO_x_ solar cells are made entirely of a metal oxide that offers an efficiency of 2.1% and which can transmit light (57%) in the visible range [5]. Such solar cells could be utilized to cover windows in buildings and produce electricity without blocking indoor space access to light since the refractive index of metal oxides is mainly between the refractive index of silicon and air. They can also be utilized as an anti-reflection coating layer for other silicon solar cell devices [6]. Furthermore, metal oxides are widely used as an electron transport layer (ETL) and hole transport layer (HTL) in perovskite-based solar cells [7]. These cells offer a novel hybrid organic and inorganic structure based on perovskite crystal for light harvesting [8].

Molybdenum trioxide MoO_3_ is an important oxide conductor with a large potential for applications in electronics due to its high oxidation state. It is proposed for utilization in the perovskite solar cell by [9] in order to improve performance. While other compounds of the low oxidation state of molybdenum are unstable and are practically not used in industry, MoO_3_ is much easier to synthesize through the Sol–Gel method [10]. MoO_3_ mainly belongs to the group of transition metal oxides, which depends on its electronic properties, size, phase state and physical–chemical properties. It exhibits good optical and electrical properties when embedded in multilayer structures or used as a functional coating. In experiments, as a result of crystallization, the optical band gap of MoO_3_ decreases from 3.1 eV to 1.7 eV [11]. Therefore, MoO_3_ can be utilized in various optoelectronic devices by controlling the band gap. There are three main polymorphic phases of MoO_3_. Due to the large band gap of MoO_3_, it cannot be used as a photocatalyst material in optoelectronics. Among them, α-MoO_3_ with an orthorhombic crystal structure is the most thermodynamically stable. Nanoscale α-MoO_3_ is widely used in electrochromic/photochromic devices [12], pseudo-volumetric charge capacitors [13], supercapacitors [14] and gas sensors [15]. Doped p-type MoO_3_ has been used as an HTL in organic solar cells [16].

Silicon-based solar cells account for 95% of solar cells produced in the industry [17]. Silicon is ubiquitous on earth and its reserves are substantial. In order to optimize silicon-based solar cells, an anti-reflection layer [18], textures [19], nanoparticles [20] and quantum dots [21] are formed on their surface and volume. The efficiency of the homojunction silicon-based solar cell does not exceed 29% according to the Shockley–Quisser limit [22]. Therefore, various silicon-based heterojunction solar cells are being designed [23]. ZnO and perovskite materials coated on the silicon surface as an emitter layer have been found to increase their efficiency [24]. On the other hand, MoO_3_ as an emitter layer for silicon has hardly been researched. Therefore, in this paper, we decided to study the MoO_3_/Si heterojunction solar cell using Sentaurus TCAD. The optoelectrical properties of MoO_3_, like ZnO, have not been widely studied. In order to model devices in Sentaurus TCAD, the physical properties of each material must be available. The first principal study determined the optical and electronic properties of MoO_3_.

## 2. Materials and Methods

Semiconductor devices are mainly simulated using TCAD programs that have a limited material database. However, it is possible to simulate devices made from materials that are not available in the TCAD program database by creating their parameter files. It is almost impossible to find all of the needed physical properties of utilized material by experimentation or simulation from the state of the art. Therefore, the physical properties of materials can be directly calculated by density functional theory (DFT). Utilizing the physical properties calculated in DFT, it is then possible to create a parameter file for utilized materials and model a device made of these materials in TCAD programs. Therefore, in this scientific work, the physical parameters of MoO_3_ were calculated using DFT and the MoO_3_/Si heterojunction solar cell was simulated in TCAD software.

### 2.1. Method of Material Simulation

Calculation of the electrical properties of MoO_3_ was carried out based on DFT using Cambridge Serial Total Energy Package (CASTEP) code [25] using on-the-fly generation (OTFG) ultrasoft, and OTFG was performed with norm-conserving pseudopotentials. A Koelling–Harmon [26] scalar relativistic correction was also utilized to account for any relativistic effects. The calculation included geometric optimization, optical properties, and band structure. Figure 1 shows the orthorhombic crystal structure of MoO_3_ with space group P21/b21/n21/m. The supercrystal consists of 12 atoms of oxygen (O) and 4 atoms of molybdenum (Mo). Before determining the properties of the crystal, it was necessary to perform geometrical optimization to find an atomic arrangement that made the structure the most stable. Crystal lattice constants and atomic positions were optimized to find the lowest energy configuration using the Broyden–Fletcher–Goldfarb–Shanno (BFGS) algorithm [27]. After geometric optimization, crystal constants were equal to a = 3.9628, b = 13.8550 and c = 3.6964.

In order to deal with exchange-correlation interaction hybrid functionals, the Perdew–Burke–Ernzerhof (PBE) functional of the generalized gradient approximation (GGA) [28] and Heyd–Scuseria–Ernzerhof (HSE06) [29] was used. The PBE functional is popular because it requires minimal computation time. However, this approach does not allow for highly accurate calculations of the band structure of metal oxides. In return, the HSE06 hybrid functional provides the most accurate results. Nevertheless, the time required for such calculations is significant. Brillouin zone structures were modeled using modified Monkhorst-Pack k-point meshes with dimensions of 6 × 2 × 7 and 4 × 1 × 4. Using the HSE06 hybrid functional allowed us to employ the OTFG norm-conserving pseudopotential on smaller 4 × 1 × 4 Monkhorst-Pack k-point grids, which considerably reduced the computational time. In the PBE functional, OTFG ultrasoft pseudopotential and 6 × 2 × 7 Monkhorst-Pack k-point grids improved the precision of the calculations, and therefore, the kinetic energy limit, the total energy, and the convergence criterion of the residual force relaxation were calculated as 571.4 eV, 1.0 × 10^−6^ eV/atom, and 0.03 eV/A, respectively. For the HSE06 hybrid functional, it was 381 eV, 1.0 × 10^−5^ eV/atom and 0.07 eV/A. All calculations were carried out in reciprocal space.

### 2.2. Method of Device Simulation

Nowadays, Lumerical [30], Silvaco [31] and Sentaurus [32] are widely used as TCAD software. In this paper, Sentaurus TCAD software was used to determine the optical and electrical properties of the MoO_3_/Si heterojunction solar cell. The heterojunction solar cell was modeled using four of them: Sentaurus Structure Editor, Sentaurus Device, Sentaurus Workbench, and Sentaurus Visual. Each instrument has an independent function. In Sentaurus Structure Editor, geometric models of devices are created along with given information about the type and concentration of doping, and the material present in each field. Additionally, the created device can be meshed with the required size. In Sentaurus Device, physical properties are assigned to each material and the necessary parameters of the device are determined using a numerical method. The results are generated in graphic and visual forms using Sentaurus Visual. Sentaurus Workbench manages every modeling task and helps every instrument work together.

The geometric model of the MoO_3_/Si heterojunction solar cell shown in Figure 2 was created by writing code in the Sentaurus Structure Editor using the Tool Command Language (TCL). The MoO_3_ layer thickness was varied from 20 nm to 1200 nm. Si layer thickness remains unchanged at 9 µm. As the MoO_3_ material tends to form n-type due to the vacancy of the oxygen atom [33], this layer was taken as n-type and the silicon base as p-type. 1 × 10^17^ cm^−3^ vacancies were assumed to exist in MoO_3_, and Boron atoms were introduced at a concentration of 1 × 10^15^ cm^−3^ to form p-type silicon. Aluminum was used as the front and back contacts of the device. A geometric model was meshed for calculation in the numerical method. Since simulation outcomes are sensitive to mesh size, the trade-off between accuracy and computation time increases as the grid size decreases. While calculation speed increases with increasing grid size, accuracy decreases dramatically. Therefore, the solar cell must be meshed at an optimal size for appropriate accuracy and speed of calculation. In this work, the relatively active heterojunction area of the solar cell was meshed with a size of 1 nm while the other regions were meshed with a size of 10 nm. Silicon is one of the most studied materials. Therefore, almost all of its physical parameters were measured in the experiment. Sentaurus TCAD’s material database also has a silicon material file. But the material properties of MoO_3_ are not available in Sentaurus TCAD. Its parameter file was formed based on the values calculated in CASTEP.

### 2.3. Theoretical Background of Device Simulation

Several methods were used in the modeling of semiconductor devices. Semiconductor devices are divided into thermoelectric, optoelectronic and electronic devices. Depending on the type of device, physical models are selected for modeling. A solar cell is an optoelectronic device, and its modeling is carried out by determining the optical and electrical properties of the utilized metal oxide. The Transfer Matrix Method (TMM) and Ray Tracing methods were utilized for optical simulation. The Ray Tracing method is mainly used to model textured solar cells [34] while multilayer planar solar cells are modeled in TMM [35] since it also takes into account the interference phenomena in the layers of solar cells. In this scientific work, since the MoO_3_/Si solar cell is planar, its optical properties were determined using TMM given in Formula (1).
(1)$$\left[\begin{array}{l} E_{i}\\ E_{r} \end{array}\right]=M\left[\begin{array}{l} E_{t}\\ 0 \end{array}\right]$$
where *M* is the matrix, *E_i_* is the electrical field of the incident light, *E_r_* is the electric field of reflected light and *E_t_* is the electric field of transmitted light.

In TMM, light absorption in each layer is calculated using the Beer–Lambert law given in Formula (2). Therefore, the concentration of absorbed photons at each point of the solar cell can be determined utilizing Formula (2).
(2)$$I=I_{0}e^{-\alpha d}$$
where *I* is the intensity, *I*_0_ is the initial intensity, *a* is the absorption coefficient of the material and *d* is the thickness of the layer.

The light beam is refracted and reflected at the boundary of two media. The relationship between the angles of refracted and incident rays is calculated using Snell’s law. Fresnel coefficients in Formula (3) were used to determine the energy balance between incoming, reflected, and absorbed photons across a boundary between two mediums [36]. These Fresnel coefficients are optical boundary conditions.
(3)$$\left\{\begin{array}{l} r_{t}=\frac{n_{1}\cos\beta -n_{2}\cos\gamma }{n_{1}\cos\beta +n_{2}\cos\gamma }\\ t_{t}=\frac{2n_{1}\cos\beta }{n_{1}\cos\beta +n_{2}\cos\gamma } \end{array}\right.\;and\;\left\{\begin{array}{l} r_{p}=\frac{n_{1}\cos\gamma -n_{2}\cos\beta }{n_{1}\cos\gamma +n_{2}\cos\beta }\\ t_{p}=\frac{2n_{1}\cos\beta }{n_{2}\cos\beta +n_{1}\cos\gamma } \end{array}\right.$$
where *r_t_* and *t_t_* are the Fresnel coefficients for transversal polarized light, *r_p_* and *t_p_* are the Fresnel coefficients for parallel polarized light, *n*_1_ and *n*_2_ are the refractive indices of first and second media, *β* is the angle of the incident light, and *γ* is the angle of refracted light.

In the reality, there is roughness on the surface of planar solar cells, therefore some of the light is scattered due to the roughness of the surface. In TMM, it is possible to calculate the scattering of light due to the surface roughness of multilayer planar structures [37]. For this, TMM should be modified. The scalar scattering theory [38] allows the calculation of the amount of light scattered on the surface. The ratio of the scattered light and the total light incident on the surface is called the haze parameter. The dependence of the scattering process on the direction was modeled using the angular distribution function [39]. The haze functions for the reflection and transmission coefficients in Formula (4) express the roughness quality of the solar cell’s surface. They adapt the matrix elements used to calculate the transmission and reflection coefficients in TMM to incorporate the scattered light as well.
(4)$$\begin{array}{l} H^{r}{}_{j,j+1}\left(\lambda ,\varphi_{j}\right)=1-\exp\left[-{\left(\frac{4\pi \sigma_{rms}c_{r}\left(\lambda ,\sigma_{rms}\right)n_{j}\cos\varphi }{\lambda }\right)}^{a^{r}}\right]\;\\ H^{t}{}_{j,j+1}\left(\lambda ,\varphi_{j}\right)=1-\exp\left[-{\left(\frac{4\pi \sigma_{rms}c_{t}\left(\lambda ,\sigma_{rms}\right)\left|n_{j}\cos\varphi_{j}-n_{j+1}\cos\varphi_{j+1}\right|}{\lambda }\right)}^{a^{t}}\right] \end{array}$$
where *σ_rms_* is the mean square roughness of the surface, and *a^r/t^* and *c_r/t_* are fitting parameters.

The AM1.5G spectrum was chosen as the light source. The concentration of generated charge carriers was determined using the quantum yield function. The quantum yield function is a logic function equal to 1 if the energy of the absorbed photon is greater than the bandgap energy of the material and it forms an electron-hole pair, otherwise, it is equal to 0, and does not form an electron-hole pair.

According to the band structure, silicon is an indirect semiconductor. According to the band structure of MoO_3_ calculated in CASTEP, it was also found to be an indirect semiconductor. The percentage of radiative recombination in indirect semiconductors is less than 1%. Therefore, radiative recombination was not taken into account in the simulation. Only Shockley–Read–Hall (SRH) and Auger recombinations were accounted for.

The electric field strength and potential around the charge carriers were calculated using the Poisson equation in Formula (5).
(5)$$\Delta \varphi =-\frac{q}{\varepsilon }\left(p-n+N_{D}+N_{A}\right)$$
where *ε* is the permittivity, *n* and *p* are the electron and hole concentrations, respectively, *N_D_* and *N_A_* are the concentrations of donor and acceptor, respectively, and *q* is the charge.

The concentration of charge carriers formed in each semiconductor is calculated using the Fermi function given in Formula (6). The concentration of charge carriers is also calculated using the Boltzmann approximation in the analysis. The error rate of the Boltzmann approximation increases when the input concentration is high. Therefore, in this scientific work, the Fermi function was used to calculate the concentration of charge carriers in each layer. The Fermi half-integral in the Fermi function can be calculated using a numerical method.
(6)$$\begin{array}{l} n=N_{c}F_{1/2}\left(\frac{E_{F,n}-E_{c}}{kT}\right)\\ p=N_{V}F_{1/2}\left(\frac{E_{V}-E_{F,p}}{kT}\right) \end{array}$$
where *N_c_* and *N_v_* are the densities of the states in the conduction and the valence bands, respectively, *E_c_* is the minimum energy of the conduction band, *E_v_* is the maximum energy of the valence band, *T* is the temperature, *k* is the Boltzmann constant, and *E_F,n_* and *E_F,p_* are the quasi-fermi energies.

Carrier transport in semiconductors creates a current. The continuity equation expresses the relationship between the change in the concentration of charge carriers in a volume of the solar cell and the current. There are four main methods for representing carrier transport [40]: drift-diffusion, thermodynamic, hydrodynamic, and Monte Carlo. The drift-diffusion model calculates the carrier transport due to the electric field strength and the difference in concentration according to Fick’s law [41]. It does not take into account temperature and other external influences on carrier transport, while thermodynamic or hydrodynamic models take into account the effect of temperature changes. In this scientific work, the drift-diffusion model given in Formula (7) was used to determine the carrier transport. Since the main goal of this work was to determine the photoelectric parameters of the MoO_3_/Si solar cell, the effect of temperature on the operation of the solar cell was not taken into account.
(7)$$\begin{array}{l} J_{n}=-nq\mu_{n}\nabla \Phi_{n}\\ J_{p}=-pq\mu_{p}\nabla \Phi_{p} \end{array}$$
where *J_n_* and *J_p_* are electron and hole currents, respectively, *μ_n_* and *μ_p_* are electron and hole mobilities, respectively, and *Φ_n_* and *Φ_p_* are the electron and hole quasi-Fermi potentials.

The electrical boundary conditions in Formula (8) represent the collection of charge carriers formed in the semiconductor at the contacts.
(8)$$\begin{array}{l} \varphi =\varphi_{F}+\frac{kT}{q}a\sinh\left(\frac{N_{D}-N_{A}}{2n_{i,eff}}\right)\;\\ n_{0}p_{0}=n_{i,eff}^{2}\\ n_{0}=\sqrt{\frac{{\left(N_{D}-N_{A}\right)}^{2}}{4}+n_{i,eff}^{2}}+\frac{N_{D}-N_{A}}{2}\\ \;p_{0}=\sqrt{\frac{{\left(N_{D}-N_{A}\right)}^{2}}{4}+n_{i,eff}^{2}}-\frac{N_{D}-N_{A}}{2} \end{array}$$
where *n_i,eff_* is the effective intrinsic carrier concentration, and *φ_F_* is the Fermi potential of the contact.

## 3. Results and Discussion

### 3.1. Band Structure

After the geometric optimization of the crystal structure, the next stage of this work was centered on band structure calculation. The band structure calculation was carried out utilizing the PBE functional and HSE06 hybrid functional. Figure 3 shows the band structure of MoO_3_ calculated using the PBE (a) and HSE06 (b) functionals. The maximum energy of the valence band corresponded to the T symmetry direction and the minimum energy of the conduction corresponded to the X symmetry direction. This satisfies the previous calculation results [42,43]. The band structure obtained with the HSE06 functional was found to be very similar to the band structure computed using the PBE functional, supporting the findings of Yu Xie’s simulation results [44]. However, the band gap energy was drastically different. In the HSE06 hybrid functional, band gap energy was found to be 3.027 eV, and in the PBE functional, it was equal to 2.12 eV.

In the experiment carried out by [45], it was found that the band gap of MoO_3_ was 3.2 eV. A 94.6% accuracy in the HSE06 hybrid functional and a 66.25% accuracy in the PBE functional were achieved by calculating the band gap of MoO_3_. Qian Qu [46] calculated the band gap of MoO_3_ using HSE06 and PBE functionals with 90% and 53.5% accuracies, respectively. Figure 4 shows the density of states of MoO_3_. There is a sharp difference in the density of the states calculated in PBE and HSE06 due to the various band gaps. In HSE06 and PBE, the density of the states at the Fermi level was equal to 2.5 electrons/eV.

### 3.2. Effective Mass

The effective mass represents carriers with different energy movements in the crystal lattice. By examining the band structure, the effective mass can be calculated. In most semiconductors, the energy range around the lowest energy of the conduction band and the highest energy of the valence band can be calculated using the parabolic function E(k) in Formula (9).
(9)$$E\left(k\right)=E_{0}+\frac{\hbar^{2}k^{2}}{2m^{*}}$$
where *E* is the energy, *E*_0_ is the energy at *k = 0*, *k* is the wavenumber, *m** is the effective mass, and ℏ is the Planck constant.

In order to determine the effective mass of the hole from the band structure given in Figure 3a, a separate set *E*(*k*) was extracted from the upper point of the valence band (Figure 5b) and points close to it. In addition, to determine the effective mass of the electron, another band structure *E*(*k*) was extracted from the lower point of the conduction band (Figure 5a) and points close to it.

Based on these points, a parabolic function of the valence band and conduction band corresponding to Formula (9) was created using the parabolic approximation of the polynomial function. In the parabolic approximation of the band structure, the effective masses of the electron and the hole were determined by the second-order differentiation of the parabolic function with respect to k, as given in Formula (10).
(10)$$m^{*}=\hbar^{2}\left(\frac{d^{2}E}{dk^{2}}\right)$$

According to the calculation results, the effective mass equals 0.06673 for an electron, and −0.10084 for a hole. So, the effective mass of the hole is 1.5 times greater than the effective mass of the electron. In Dandogbessi’s work [47], the effective mass of the hole was found to be three times larger. It was determined that the probability of tunneling in MoO_3_ [48] is high due to the very light-effective masses of electrons and holes.

### 3.3. Carriers Mobility

One of the primary kinetic characteristics of charge carriers in a semiconductor is mobility. In materials science, the mobility of charge carriers is mainly calculated using Boltzmann Transport Theory (BTT) [49]. BTT cannot be calculated using the CASTEP code of the first-principle method. Therefore, a new method for BTT is found and calculated in another way. However, using the theory of Shockley and Barden [50], it is possible to determine the mobility of charge carriers only by calculating the band structure and the effect of mechanical force. Electron and hole mobilities are calculated using Formula (11) [51].
(11)$$\mu =\frac{{\left(8\pi \right)}^{\frac{1}{2}}h^{4}eC_{ij}}{3{\left(m^{*}\right)}^{\frac{5}{2}}{\left(k_{b}T\right)}^{\frac{3}{2}}E_{ij}^{2}}$$
where *e* is the electron charge, *T* is the absolute temperature, *k_b_* is the Boltzmann constant, *C_ij_* is the elastic constant of the crystal, and *E_ij_* is the energy change per unit volume.

Since the monocrystalline silicon grown in a 111 direction was selected as the base in the MoO_3_/Si solar cell, a band structure was calculated to determine the elastic modulus by applying a mechanical force to the crystal in the (1,1,1) direction. Furthermore, MoO_3_ grown on silicon should have (1,1,1) orientations. Therefore, in order to determine the mobility of charge carriers in the (1,1,1) direction of MoO_3_, the force was applied in this direction. According to the obtained result, it was determined that the elastic modulus C was equal to 184 GPa. The volume of the crystal in the initial state was *V_0_* = 202.949341 A^3^, and after applying the mechanical force it was equal to V = 202.340036 A^3^. In addition, in the initial state, the maximum energy of the valence band was *E_V0_*-*E_F_* = −0.00531 eV and the maximum energy of the conduction band was *E_C0_*-*E_F_* = 2.18 eV. After applying the force, it was equal to *E_V_*-*E_F_* = −0.06005 eV and *E_C_*-*E_F_* = 2.24 eV, respectively. The change in the valence band maximum or conduction band minimum energy per unit volume was calculated using Formula (12) [50].
(12)$$E_{ij}=\frac{dE}{\left(\frac{dV}{V_{0}}\right)}$$
where *dE* is the energy difference before and after stress, *V*_0_ is the initial volume of the crystal, and *dV* is the volume difference before and after stress.

According to the results obtained after applying the mechanical force, the change in the maximum energy of the valence band per unit volume is *E_ij_*^(v^) = 18.247 eV, and the change in the minimum energy of the conduction band per unit volume is *E_ij_*^(c)^ = 160 eV. According to the parameters determined using the PBE functional, the electron and hole mobilities were also calculated and equal to 3811.11 cm^2^V^−1^s^−1^ and 1630.39 cm^2^V^−1^s^−1^, respectively.

### 3.4. Optical Properties

It is vital to know the optical properties of materials that are used in optoelectronic devices. The most basic optical property of materials is the complex refractive index given in Formula (13).
(13)N=n+ik
where *n* is the real part and *k* is the imaginary part of the complex refractive index.

The experiment measures the absorption and reflection coefficients to determine the material’s complex refractive index. The system given in Formula (14) is solved using the experimentally measured absorption and reflection coefficients, and the real and imaginary parts of the complex refractive index are determined.
(14)$$\left\{\begin{array}{l} A=1-\exp\left(-\frac{4\pi kx}{\lambda }\right)\\ R=\frac{{\left(n-1\right)}^{2}+k^{2}}{{\left(n+1\right)}^{2}+k^{2}} \end{array}\right.$$
where *x* is the position, λ is the wavelength, *R* is the reflection coefficient, and *A* is the absorption coefficient.

The interaction of photons and electrons in the system is described using the terms of excitation of the main electrons depending on time. There are two main methods for calculating optical properties: DFT Khon–Sham orbitals [52] and time-dependent DFT (TD-DFT) [53]. TD-DFT calculation accuracy is high but its computation requires a lot of time. The imaginary part of the complex permittivity of MoO_3_ was calculated using DFT Khon–Sham orbital theory. It can be considered that the complex permittivity describes the real transitions between occupied and unoccupied electronic states. The real and imaginary parts of the complex permittivity are linked by the Kramers–Kronig [54] relationship. This relationship was used to determine the real part of the complex permittivity. The calculated results show the dependence of complex permittivity on the light wavelength, as shown in Figure 6. The real part of the complex permittivity represents the relative permittivity. Furthermore, the relative permittivity at high frequency represents the optical permittivity. According to the obtained results, it was determined that the optical permittivity is equal to ε = 58.

After calculating the real and imaginary parts of the complex permittivity, the real and imaginary parts of the complex refractive index were determined by solving the system of equations given in Formula (15). Based on the obtained results, the dependence of the real and imaginary parts of the complex refractive index on the light wavelength is shown in Figure 7.
(15)$$\left\{\begin{array}{c} \varepsilon_{1}=n^{2}-k^{2}\\ \varepsilon_{2}=2nk \end{array}\right.$$
where *ε*_1_ is the real part and *ε*_2_ is the imaginary part of the complex permittivity.

In the ultraviolet spectrum, the imaginary part of the complex refractive index of MoO_3_ is equal to *k* = 2, which means that the absorption coefficient equals 8.37 × 10^7^. It absorbs mainly ultraviolet light as other metal oxides [55]. As the wavelength increased from 100 nm to 330 nm, the refractive index significantly increased from 0.5 to 3.2. Like other metal oxides, the real and imaginary parts of the complex refractive index decreased with increasing wavelength.

### 3.5. I-V Characteristics

The I-V characteristics of a solar cell must be measured to ascertain its photoelectric parameters. In an experiment, the I-V characteristic is determined by measuring the relationship between the current and the voltage generated when the solar cell is illuminated using a variable resistance. Figure 8 shows the I-V characteristics of MoO_3_/Si solar cells with MoO_3_ layer thicknesses of 20 nm, 100 nm and 1000 nm. When the thickness of the MoO_3_ layer changed, the output power and short-circuit current changed, but the open circuit voltage did not change. According to the I-V characteristics obtained in the simulation, the short-circuit current of the MoO_3_/Si solar cell did not increase linearly depending on the thickness of the MoO_3_ layer. The short-circuit current increased when the thickness changed from 20 nm to 100 nm, and decreased when the thickness changed from 100 nm to 1000 nm. The efficiency of the MoO_3_/Si heterojunction solar cell with a MoO_3_ layer thickness of 100 nm and Si layer thickness of 9 μm was 8.8%. In the simulation, the efficiency of a silicon-based homojunction solar cell with the same size was equal to 7.56%. It means that the efficiency of the MoO_3_/Si solar cell is 1.16 times higher than that of a homojunction silicon solar cell of the same size. Due to the flat surface and a very thin base thickness, we obtained a low efficiency for n-MoO3/p-Si and n-Si/p-Si with the same sizes. In [34], it was found that the efficiency of the silicon homojunction solar cell can reach 21% if its surface was textured and the base thickness was 190 μm. Therefore, if the surface of the MoO3/Si solar cell was textured and the base thickness was more than 190 μm, its efficiency could reach 24.4%. According to the results obtained with CASTEP, it was determined that MoO_3_ mainly absorbs ultraviolet rays in the range of 100–300 nm. Further, silicon mainly absorbs rays with a wavelength in the range of 300–800 nm in the visible field. In addition, according to the result given in Figure 7, the refractive index of MoO_3_ is an average of 2.5 in the visible spectrum region. The refractive index of silicon is 3.88 [56]. Therefore, since the refractive index of MoO_3_ is between the refractive indices of air and silicon, it also acts as an anti-reflection layer [57] in silicon-based solar cells. In addition, it serves to expand the absorption spectrum of the silicon-based solar cell due to its good absorption of rays in the ultraviolet range. Therefore, the efficiency of the MoO_3_/Si heterojunction solar cell was 1.24% higher than that of the silicon-based homojunction solar cell.

According to the I-V characteristics given in Figure 8, since the photoelectric parameters change nonlinearly depending on the thickness, the dependence of the short-circuit current on the thickness of the MoO_3_ layer was studied and shown in Figure 9. The MoO_3_ layer thickness was changed from 20 nm to 1200 nm. A sharp change in short-circuit current was observed in the thin layers. When the thickness was 20 nm, the minimum short-circuit current was 15.54 mA/cm^2^; when the thickness was 60 nm, the maximum short-circuit current was 18.15 mA/cm^2^.

When the thickness changed from 60 nm to 160 nm, the short-circuit current decreased by 3.48 mA/cm^2^, and when the thickness changed from 160 nm to 200 nm, the short-circuit current increased by 0.6 mA/cm^2^. When the thickness increased from 200 nm to 1200 nm, the short-circuit current decreased linearly to 1.41 mA/cm^2^. When the thickness of MoO_3_ was thin, the amount of absorption in the emitter layer was very small but high in the base. Figure 10 shows the dependence of the absorption coefficient of the MoO_3_/Si solar cell on the wavelength of light when the thickness of the MoO_3_ layer was 20 nm, 60 nm and 160 nm. When the thickness of the MoO_3_ layer was 60 nm, the absorption coefficient in the visible region of the spectrum was the highest. Therefore, MoO_3_ with a thickness of 60 nm was the optimal anti-reflection layer for silicon. Thus, the short-circuit current of the solar cell was the highest at this thickness. Since MoO_3_ has a band gap of 3.2 eV, it mainly absorbs short-wavelength light. Hence, the maximum absorption shifts to the UV region when the thickness increases. When the thickness of the MoO_3_ layer increases, the amount of light absorption in this layer increases, but its anti-reflection property decreases. Therefore, the amount of light absorption in the silicon base decreases. Since silicon mainly absorbs rays in the visible range [58], the absorption coefficient in the visible range decreases with increasing thickness.

## 4. Conclusions

The use of metal oxides in photovoltaics as anti-reflection layers and transparent electrically conductive electrodes is becoming popular. The main purpose of this scientific work was to calculate the physical parameters of MoO_3_, research the MoO_3_/Si heterojunction solar cell and determine the optimal thickness for the MoO_3_ layer. For that, the physical parameters of MoO_3_ were determined using the CASTEP code. PBE and HSE06 functionals were used to calculate the electronic properties of MoO_3_. In both HSE06 and PBE, MoO_3_ was found to be an indirect semiconductor. However, the band gap was calculated with 94% accuracy in HSE06 and 66.25% accuracy in PBE. Therefore, it is recommended to use HSE06 when calculating the electronic properties of metal oxides. According to the obtained results, the band gap and electron affinity of MoO_3_ proved that it could be an n-type semiconductor. Therefore, it was predicted that MoO_3_ could be used as an emitter layer for a silicon-based solar cell, and a MoO_3_/Si heterojunction solar cell was investigated using Sentaurus TCAD. Since the bandgap and electron affinity of silicon and MoO_3_ are acceptable for the formation of a high-quality heterojunction, there were no disturbances in the I-V characteristics of the MoO_3_/Si structure. According to the results obtained using TCAD, it is possible to use MoO_3_ as both the anti-reflective layer and the emitter layer for the silicon-based solar cell. In addition, since the electron mobility in MoO_3_ is three times greater than that of the hole, it can be used as an electron transport layer for a perovskite-based solar cell.

## Figures and Tables

**Figure 1 nanomaterials-12-04240-f001:**
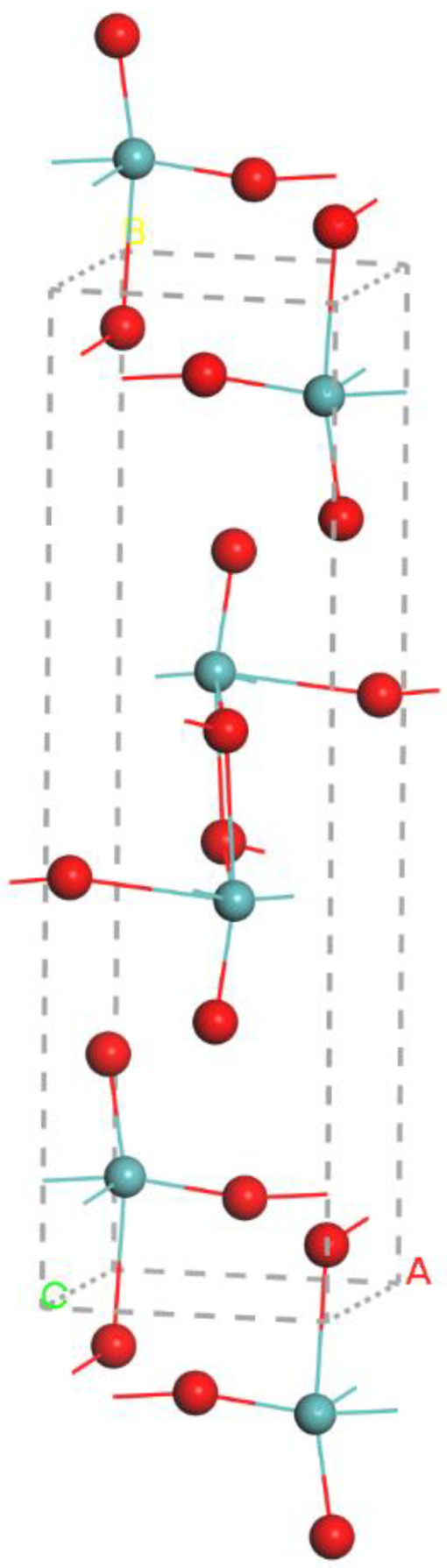
Crystal structure of MoO_3_.

**Figure 2 nanomaterials-12-04240-f002:**
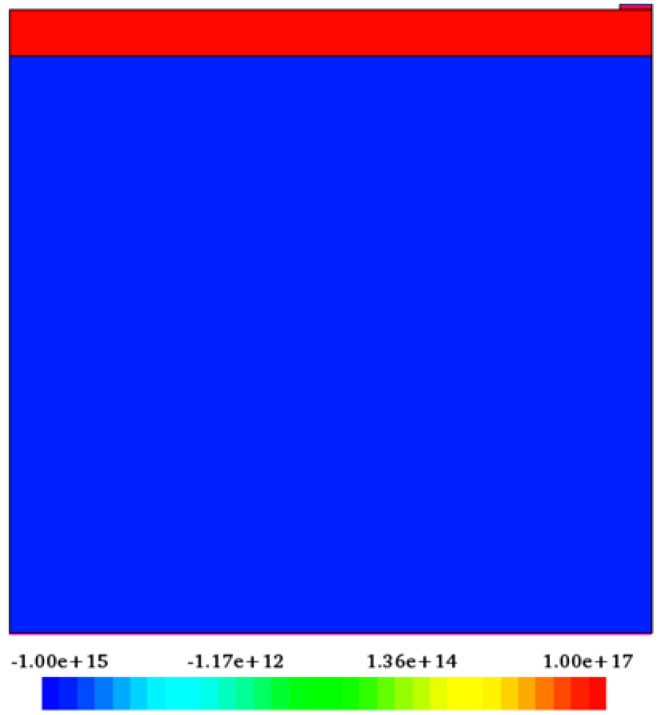
Geometrical model of MoO_3_/Si heterojunction solar cell.

**Figure 3 nanomaterials-12-04240-f003:**
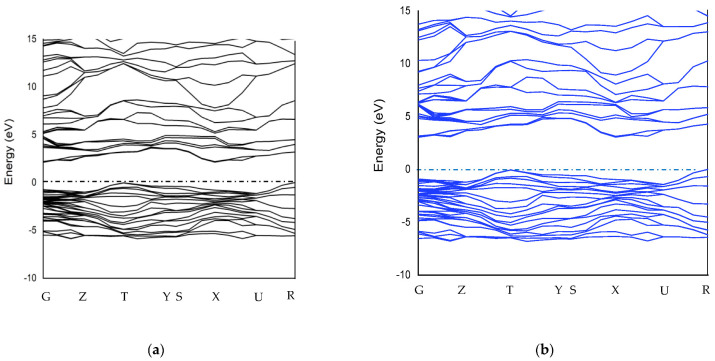
Band structure of MoO_3_ calculated using PBE functional (**a**) and HSE06 hybrid functional (**b**).

**Figure 4 nanomaterials-12-04240-f004:**
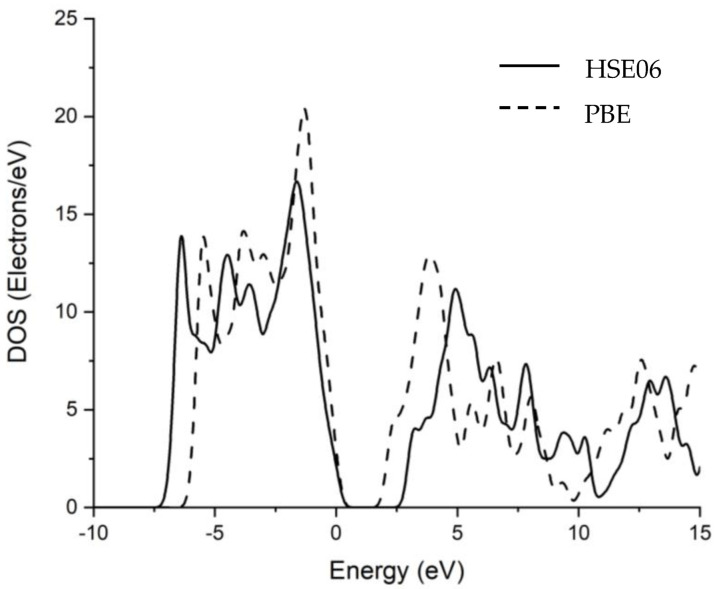
Density of States of MoO_3_ calculated using PBE and HSE06.

**Figure 5 nanomaterials-12-04240-f005:**
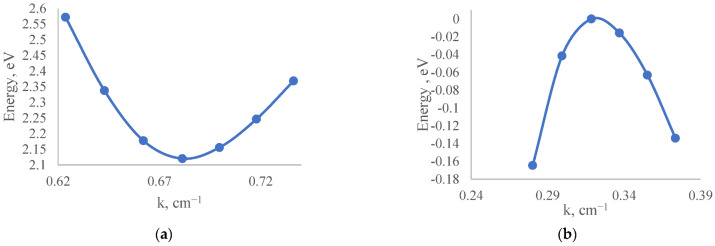
Band structure around the conduction band minima (**a**) and valence band maxima (**b**).

**Figure 6 nanomaterials-12-04240-f006:**
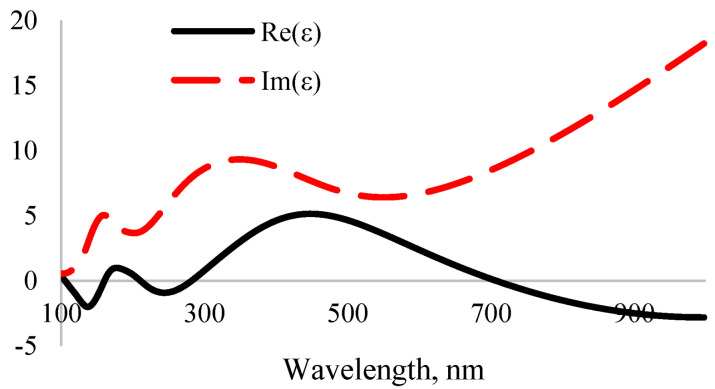
Dependence of the complex permittivity of MoO_3_ on light wavelength.

**Figure 7 nanomaterials-12-04240-f007:**
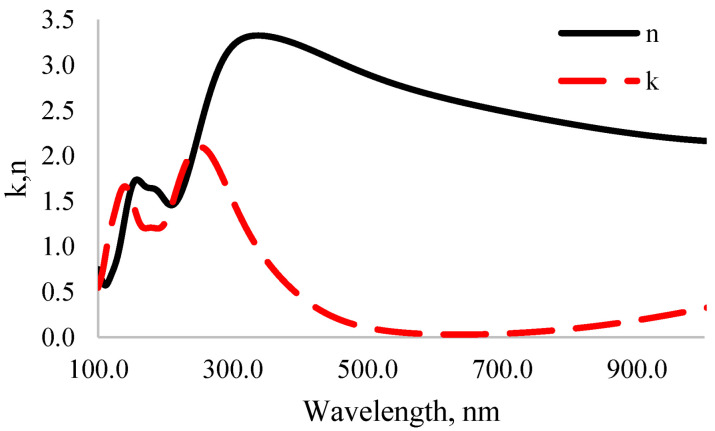
Dependence of the complex refractive index of MoO_3_ on light wavelength.

**Figure 8 nanomaterials-12-04240-f008:**
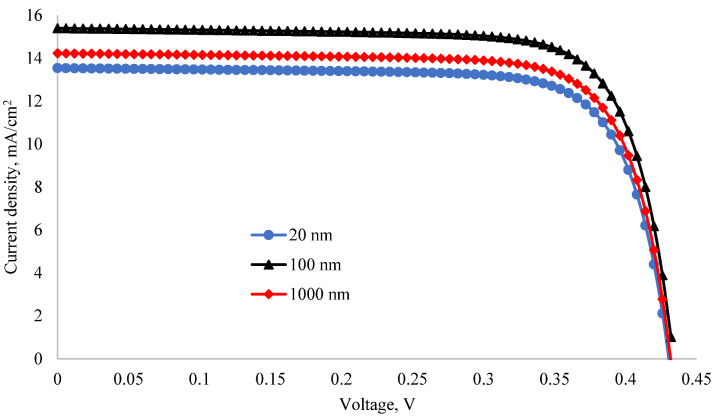
I-V characteristic of the MoO_3_/Si solar cell with different thicknesses of MoO_3_.

**Figure 9 nanomaterials-12-04240-f009:**
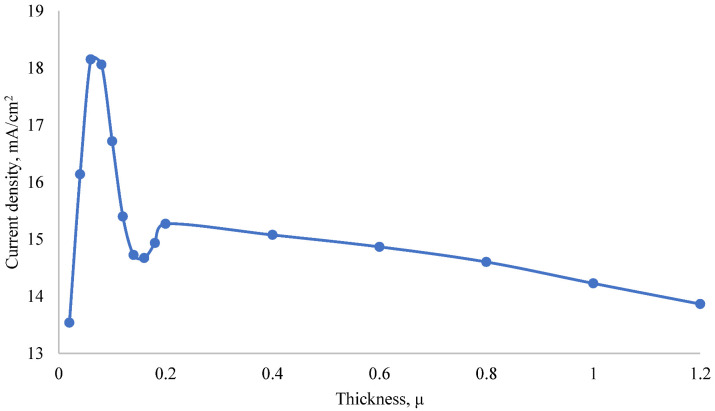
Dependence of the short-circuit current of the MoO_3_/Si heterojunction solar cell on the thickness of the MoO_3_ layer.

**Figure 10 nanomaterials-12-04240-f010:**
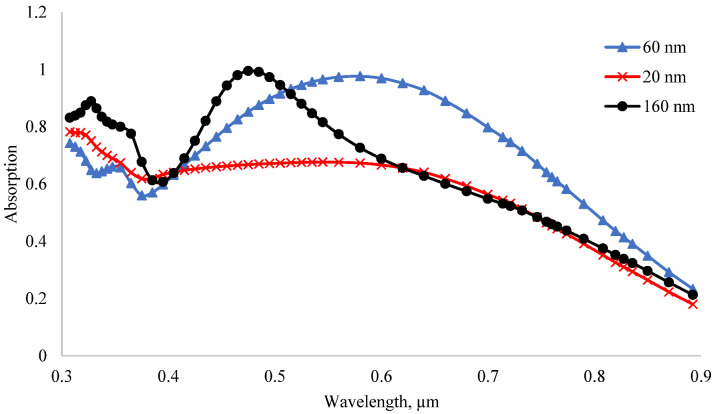
Absorption coefficient of MoO_3_/Si heterojunction solar cells.

## Data Availability

Data will be made available upon request from the corresponding author.

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
