# Peer review of "Atom-to-Device Simulation of MoO3/Si Heterojunction Solar Cell"

_nanomaterials, 2022, doi:10.3390/nano12234240_

Round 1
Reviewer 1 Report
This work carefully calculates the physical parameters of MoO3, study the MoO3/Si heterojunction solar cells and determine the optimal thickness of the MoO3 layer. Reasonable calculations processes and results have been discussed. However, this novelty of this work should be emphasized. I recommend this manuscript to be accepted subject to major revisions.
1. The materials properties of MoO3 have been extensively studied via experiments. This work does not afford deeper understanding of this material. The authors should more about the significance of this work.
2. The obtained optimal MoO3/Si solar cell shows a low efficiency of 8.8%, far below that of devices by using other emitter layer based silicon solar cells. If the theoretical efficiencies is only 8.8%, this device structure would show no future prospects for applications. The authors are suggested to compare the results with corresponding experimental results.
3. The English of this manuscript requires significant improvements.
4. There are some typos in the main text. For example, in the abstract part: “… and a Si layer thickness of 9 m is 8.8%”.
Author Response
- The materials properties of MoO3 have been extensively studied via experiments. This work does not afford deeper understanding of this material. The authors should more about the significance of this work.
Answer: Indeed, properties of MoO3 are explored studied. However, before doing this research, our goal was only to study of MoO3/Si heterojunction using TCAD software. We know that to simulate devices in TCAD, optic and electronic properties of each of the materials is necessary. There isn’t material file for MoO3 in material database of Sentaurus TCAD. Thus, we tried to collect necessary optic and electric properties of MoO3 from literatures related to experimentations. Since 2010, a lot of researches on MoO3 were related to DFT and 2D layered structures not bulk and experiment. Therefore, we decided to study properties of bulk MoO3 using DFT in order to research MoO3/Si heterojunction solar cell in Senaturus TCAD and create atom to device simulation chain. All properties of MoO3 obtained our research were discussed scientifically and compared to experimental results. In order to stay coherent and to not make any exaggeration we modified the end of introduction section as well as deleted the following sentence “The optoelectrical properties of MoO3, like ZnO, have not been widely studied.”
- The obtained optimal MoO3/Si solar cell shows a low efficiency of 8.8%, far below that of devices by using other emitter layer based silicon solar cells. If the theoretical efficiencies is only 8.8%, this device structure would show no future prospects for applications. The authors are suggested to compare the results with corresponding experimental results.
Answer: The efficiency of the MoO3/Si heterojunction solar cell with MoO3 layer thickness of 100 nm and Si layer thickness of 9 μm was 8.8%. In the simulation, the efficiency of a silicon-based homojunction solar cell with the same size was equal to 7.56%. In this article we tried to show the future potential of MoO3/Si structure by comparing it with homojunction silicon solar cell with same sizes. To show the future potential of MoO3/Si, we added some information to discussion: “It means that the efficiency of MoO3/silicon solar cell is 1.16 times higher than that of homojunction silicon solar cell with the same size. Due to flat surface and very thin thickness of base, we obtained low efficiency for n-MoO3/p-Si and n-Si/p-Si with same sizes. In [37] work, it was found that efficiency of silicon homojunction solar cell can reach 21% if its surface was textured and thickness of base was 190 μm. So, if surface of MoO3/Si solar cell will be textured and thickness of base will be more than 190 μm, its efficiency can reach 24.4%.”
- The English of this manuscript requires significant improvements.
Answer: The English of article was improved significantly.
- There are some typos in the main text. For example, in the abstract part: “… and a Si layer thickness of 9 m is 8.8%”.
Answer: Typos were corrected. All corrections were highlighted in yellow.
Reviewer 2 Report
Reviewer report on Manuscript Draft ‘Atom-to-device simulation of MoO3/Si heterojunction solar cell’.
According to the results obtained using Technology Computer Aided Design (TCAD) performed in this research, it is possible to use MoO3 as both the anti-reflective layer and the emitter layer for the silicon-based solar cell. In addition, since the electron mobility in MoO3 is three times greater than that of the hole, it can be used as an electron transport layer for a perovskite-based solar cell.
This investigation is very interesting from the point of view of materials chemistry and it is well presented in the manuscript. The research is well conducted and well addressed in manuscript, it is in the scope of the journal. Therefore, the manuscript can be published after some minor improvements:
All abbreviations should be explained in the text, e.g. TCAD is not explained, Technology Computer Aided Design (TCAD), etc.
Other application areas for here assessed heterojunction, e.g. in the design of gas sensors (Review Insights in the Application of Stoichiometric and Non-Stoichiometric Titanium Oxides for the Design of Sensors for the Determination of Gases and VOCs (TiO2−x and TinO2n−1 vs. TiO2). Sensors 2020, 20, 6833.) could be predicted, overviewed and discussed in discussion and conclusion parts of the manuscript.
Author Response
This investigation is very interesting from the point of view of materials chemistry and it is well presented in the manuscript. The research is well conducted and well addressed in manuscript, it is in the scope of the journal. Therefore, the manuscript can be published after some minor improvements:
All abbreviations should be explained in the text, e.g. TCAD is not explained, Technology Computer Aided Design (TCAD), etc.
Answer: All abbreviations were explained before using them in text.
Other application areas for here assessed heterojunction, e.g. in the design of gas sensors (Review Insights in the Application of Stoichiometric and Non-Stoichiometric Titanium Oxides for the Design of Sensors for the Determination of Gases and VOCs (TiO2−x and TinO2n−1 vs. TiO2). Sensors 2020, 20, 6833.) could be predicted, overviewed and discussed in discussion and conclusion parts of the manuscript.
Answer: We added this reference to discuss future prospect of the MoO3.
Reviewer 3 Report
In this work the authors calculated the photoelectric parameters of the MoO3/Si hetero-junction solar cell using Sentaurus TCAD. According to the obtained results, the efficiency of the MoO3/Si solar cell with a MoO3 layer thickness of 100 nm and a Si layer thickness of 9 m is 8.8%, which is 1.24% greater than the efficiency of a homojunction silicon-based solar cell of the same size.
This is an interesting work; Nevertheless revisions are needed in order to publish this work to MDPI Nano materials.
1. Several syntax errors need to be corrected.
Starting from the abstract of the manuscript, the authors state: ..calculated the photoelectric parameters of the MoO3/Si hetero-junction solar cell using Sentaurus TCAD, this..
"This" should be erased.
Several other syntax errors should be corrected..
2. Sentaurus Device is an advanced multidimensional device simulator capable of simulating electrical, thermal, and optical characteristics of silicon-based and compound semiconductor devices, for designing and optimizing current and future semiconductor devices.
The authors shout not use the commercial name of the device in the abstract. Use a description instead. People who are not familiar with the technique do not know the name or the terms..
3. The authors could add more references in the results and discussion part of the their manuscript.
4. I suggest the authors to discuss further their figures. It seems that further analysis is needed.
Author Response
This is an interesting work; Nevertheless revisions are needed in order to publish this work to MDPI Nano materials.
- Several syntax errors need to be corrected.
Starting from the abstract of the manuscript, the authors state: calculated the photoelectric parameters of the MoO3/Si hetero-junction solar cell using Sentaurus TCAD, this.
"This" should be erased.
Several other syntax errors should be corrected.
Answer: Syntax errors were corrected. All corrections were highlighted in yellow.
- Sentaurus Device is an advanced multidimensional device simulator capable of simulating electrical, thermal, and optical characteristics of silicon-based and compound semiconductor devices, for designing and optimizing current and future semiconductor devices.
The authors shout not use the commercial name of the device in the abstract. Use a description instead. People who are not familiar with the technique do not know the name or the terms.
Answer: Abstract was modified according to your suggestions.
- The authors could add more references in the results and discussion part of the their manuscript.
Answer: Discussion part of article was extended by adding some references.
- I suggest the authors to discuss further their figures. It seems that further analysis is needed.
Answer: Figures were discussed properly according to your suggestions.
Round 2
Reviewer 1 Report
Accept.
Reviewer 3 Report
The authors have revised their manuscript following my comments.
This work could be published in its present form.